# ConciseHint: Boosting Efficient Reasoning via Continuous Concise Hints during Generation

## Abstract

Recent advancements in large reasoning models (LRMs) like DeepSeek-R1 and OpenAI o1 series have achieved notable performance enhancements on complex reasoning tasks by scaling up the generation length by Chain-of-Thought (CoT). However, a critical issue is their tendency to produce excessively verbose reasoning processes, leading to the inefficiency problem. Existing literature on improving efficiency mainly adheres to the before-reasoning paradigms such as prompting and reasoning or fine-tuning and reasoning, but ignores the promising direction of directly encouraging the model to speak concisely by intervening during the generation of reasoning. In order to fill the blank, we propose a framework dubbed ConciseHint, which continuously encourages the reasoning model to speak concisely by injecting learnable hints (manually designed or learned on concise data) during the generation of the reasoning. Besides, ConciseHint is adaptive to the complexity of the query by adaptively adjusting the hint intensity, which ensures it will not undermine model performance. Experiments on the state-of-the-art LRMs, including DeepSeek-R1 and Qwen-3 series, demonstrate that our method can effectively produce concise reasoning while maintaining the performance well. Moreover, we show that ConciseHint is flexible and can be seamlessly integrated with existing methods to further push the upper bound of the efficiency.

## 1 Introduction

Reasoning ability is significant for large language models (LLMs) (Liu et al., 2024; Yang et al., 2024; Grattafiori et al., 2024; Hurst et al., 2024; Ouyang et al., 2022) to execute effectively across a wide range of complex tasks (Zhao et al., 2023; Chang et al., 2024; Qu et al., 2025; Hao et al., 2023; Wei et al., 2022), including arithmetic reasoning, commonsense reasoning, etc. Chain of thought (Wei et al., 2022; Kojima et al., 2022) (CoT) is the most popular manner to enhance the reasoning ability for LLMs by explicitly generating intermediate reasoning steps. Recently, state-of-the-art reasoning models (e.g., Gemini-2.5 (Deepmind, 2025), OpenAI-o1 (Jaech et al., 2024) and DeepSeek-R1 (Guo et al., 2025)) have internalized the chain-of-thought paradigm instead of few-shot (Wei et al., 2022) or zero-shot prompting (Kojima et al., 2022).

Although large reasoning models (LRMs) with CoT demonstrate remarkable performance, a critical limitation lies in the inefficiency of their reasoning process (Qu et al., 2025; Liu et al., 2025; Sui et al., 2025; Feng et al., 2025; Han et al., 2024). Typically, the output of reasoning models consists of far more tokens compared to common LLMs, due to the detailed and usually verbose intermediate reasoning steps, leading to substantial computational costs and high inference latency. For example, LRMs usually present unnecessary coherence tokens (Su et al., 2025) or perform redundant self-checks (Qu et al., 2025; Fu et al., 2025).

To improve the efficiency by making the reasoning model speak more concisely, mainstream methods follow the two paradigms: (i) Prompting in the input stage: Adding extra control prompts (Renze & Guven, 2024; Han et al., 2024; Lee et al., 2025; Aytes et al., 2025) like "Be concise." to the model at the input stage, and then perform the reasoning. (2) Finetune-and-use: Internalizing the conciseness by optimizing the model with supervised fine-tuning (SFT) (Xia et al., 2025; Munkhbat et al., 2025; Ma et al., 2025) or reinforcement learning (RL) (Shen et al., 2025; Luo et al., 2025), and then perform the reasoning. They don't directly intervene during the reasoning stage when the model generates tokens one by one. Therefore, an orthogonal and largely unexplored question arises: Is it possible

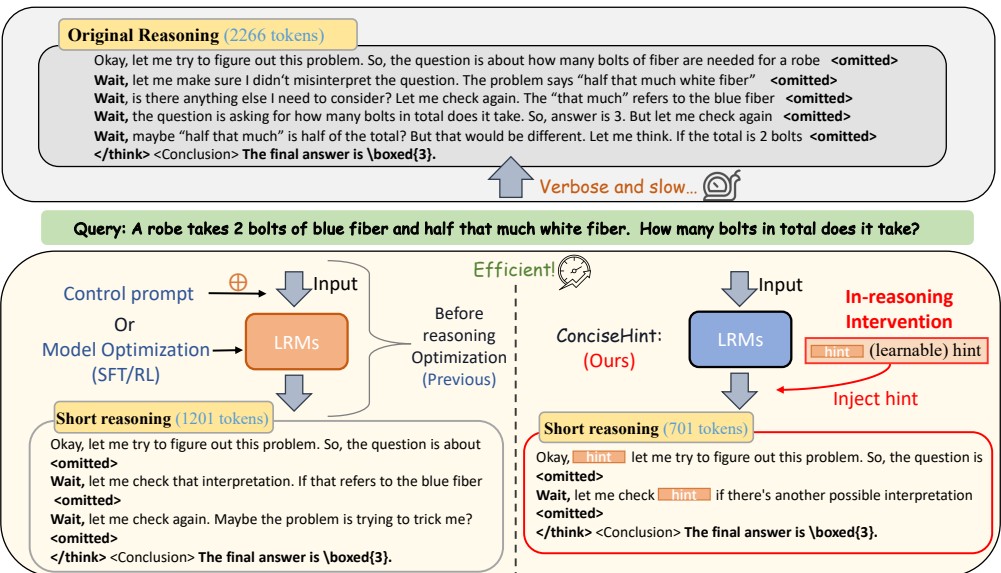

Figure 1: Previous works mainly enhance conciseness before the actual reasoning (i.e., adding the control prompt or optimizing the model via SFT/RL), while we focus on intervening during the reasoning process to encourage conciseness, i.e., in-reasoning intervention. ConciseHint achieves this goal by continuously injecting learnable hints during the generation.

to guide the reasoning model to speak more concisely by intervening during the generation of the intermediate reasoning steps? We point out two key points needed to answer this question: one is to design an approach to enable effective intervention during reasoning, and the other is to select an optimal intensity of intervention adaptively to the complexity of a given query.

To fill the blank, we propose ConciseHint, which performs intervention during the generation of reasoning, encouraging the model to speak concisely by injecting hints, as illustrated in Figure 1. Specifically, ConciseHint continuously influences the reasoning by injecting the hint that can either be a manually designed text (e.g., "make answer concise!") or continuous embeddings learned on a concise dataset. Both types of hints can encourage the subsequent token generation to be more concise. Trained on concise data, the learned hint can capture concise patterns inherent in the data, thereby further enhancing the efficiency over the manual hint. Besides, the controllability of the reasoning length can be easily achieved by interpolating in the embedding space. Additionally, ConciseHint adaptively adjust the injection intensity according to the complexity of the query, as easy queries can usually tolerate a larger compression ratio of reasoning than complex ones. This complexity-adaptive strategy facilitates a good efficiency-accuracy balance by employing a lower hint intensity for complex queries and a higher intensity for easy ones. Moreover, ConciseHint dynamically adjusts the position of the injection to ensure a good computing-accuracy balance.

To evaluate ConciseHint, we conduct experiments on the state-of-the-art large reasoning models (DeepSeek-R1 (Guo et al., 2025) and Qwen-3 (Alibaba, 2025) series) with a range of benchmarks (AIME24, GSM8K, and GPQA-Diamond) with varying complexity levels. Experimental results indicate that our in-reasoning intervention framework can effectively improve the reasoning efficiency while maintaining the model performance well. Moreover, they also demonstrate that ConciseHint can serve as a flexible plugin that can be seamlessly integrated with existing methods to further enhance the efficiency, effectively pushing the upper bound of the efficiency.

## 2 RELATED WORKS

### 2.1 REASONING MODELS AND THE INEFFICIENCY ISSUE

The emergence of chain-of-thought (Wei et al., 2022; Kojima et al., 2022) endowed LLMs with powerful reasoning ability through explicitly generating intermediate reasoning steps. Initially,

vanilla LLMs such as GPT-4o (Hurst et al., 2024) and PaLM (Chowdhery et al., 2023) can obtain the enhanced reasoning ability by few-shot (Wei et al., 2022) or zero-shot (Kojima et al., 2022) prompting. Recently, large reasoning models such as Gemini-2.5 (Deepmind, 2025), OpenAI-o1 (Jaech et al., 2024) and DeepSeek-R1 (Guo et al., 2025) have internalized the reasoning ability by supervised fine-tuning (SFT) and reinforcement learning (RL), no longer needing manual prompting. While demonstrating superior performance compared to common LLMs, reasoning models incur high computational costs due to the detailed and usually verbose reasoning process (Qu et al., 2025; Zhao et al., 2023; Chang et al., 2024), leading to inefficiency in reasoning. For example, substantial works point out that reasoning models usually overthink simple queries (Qu et al., 2025; Chen et al., 2024; Shen et al., 2025), generate verbose multiple rounds of self-check (Qu et al., 2025; Fu et al., 2025), and allocate a substantial proportion of tokens to support textual coherence (Su et al., 2025) rather than the core reasoning advancement. These sorts of inefficiency issues result in the waste of computational resources and energy.

## 2.2 EFFICIENT METHODS FOR REASONING MODELS

Recently, researchers have paid attention to alleviating the inefficiency of large reasoning models. Existing methods can be roughly divided into three groups (Qu et al., 2025), i.e., training-free methods, SFT-based methods, and RL-based methods. SFT-based methods either fine-tune the reasoning model to internalize the concise reasoning patterns on the curated concise datasets (Xia et al., 2025; Munkhbat et al., 2025), or replace explicit token generation in the reasoning process by predicting answers based on internal latent representations (Deng et al., 2024; Hao et al., 2024). RL-based methods usually incorporate the length constraint into the reward function to encourage conciseness (Shen et al., 2025; Luo et al., 2025), or teach the model "when to think" (Huang et al., 2025; Fang et al., 2025; Zhang et al., 2025a). In contrast, training-free methods do not involve training, which is easy to use and can serve as a plug-in. For example, prompt-based methods (Renze & Guven, 2024; Han et al., 2024; Aytes et al., 2025) add control prompts to the user input to encourage answering concisely. Early exit methods (Fu et al., 2025; Yang et al., 2025) terminate the thinking in advance when meeting certain confidence conditions.

The previous literature mainly conforms to the paradigm of prompting or optimizing the model before using it to perform reasoning generation, and does not dynamically intervene in the model during the token generation for reasoning to make it speak more concisely. In this work, we aim to explore whether we can enhance the conciseness by continuously exerting influence during the reasoning.

## 3 THE PROPOSED CONCISEHINT FRAMEWORK

In this section, we elaborate on our proposed ConciseHint that encourages models to speak concisely by continuously and adaptively exerting influence on the reasoning process. ConciseHint injects learnable hints into the reasoning process to enhance efficiency. To avoid excessive intervention in complex queries while maintaining intensive intervention for easy queries, ConciseHint adaptively controls the injection intensity, ensuring it is negatively correlated with the complexity. To avoid compromising accuracy and achieve computational savings, ConciseHint dynamically determines the injection position, from head to tail progressively. Both manual and learned hints can encourage the subsequent reasoning to be more concise. Even though the manual hint (denoted as ConciseHint) can already achieve significant efficiency improvement in a training-free way, the learned hint trained on concise data (denoted as ConciseHint-T) can further enhance the efficiency by capturing concise patterns inherent in the data. Controllability of the reasoning length can be easily achieved by interpolating in the embedding space. The overall framework is presented in Figure 2.

**ConciseHint continuously injects the hint in a complexity-adaptive way.** Specifically, ConciseHint continuously injects the hint like "make answer concise!" in the reasoning process. For instance, if the original text is "Okay, let me try to figure out this problem. The problem says a robe takes 2 bolts of blue fiber and half that much white fiber" will be modified to "Okay, make answer concise! let me try to figure out this problem. The problem says a robe takes 2 bolts of blue fiber and half that much white fiber". Injecting the hint can encourage the following reasoning to be more concise. However, a critical problem is how to select an optimal injection intensity for a given query. An excessively high injection intensity will harm the accuracy, particularly for complex queries, while a low intensity will decrease the efficiency improvement (see Table 3 in the ablation study). We propose to tackle this

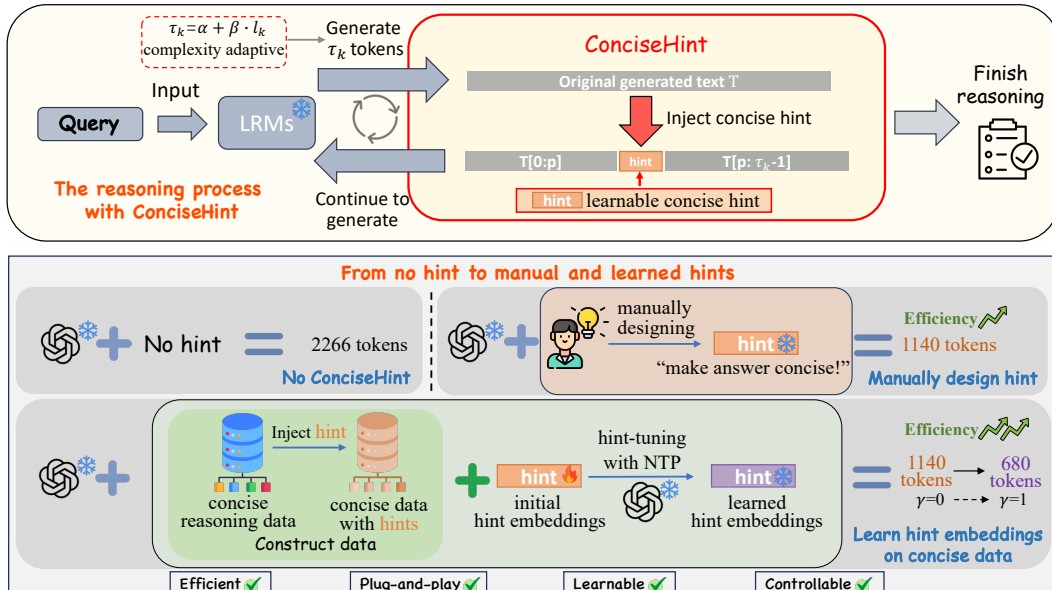

Figure 2: The illustration of ConciseHint(-T) framework. Upon obtaining $\tau_k$, the LRM generates the next $\tau_k$ tokens, injects the hint, and updates $l_k$ and $\tau_k$ in sequence, repeating this cycle until the reasoning is finished. The corresponding pseudo-code is shown in Algorithm 1. There are two ways of obtaining the hint. Firstly, we can manually design the text with expertise and prior knowledge. Secondly, we can train the hint embeddings on concise reasoning data with SFT in a Next-token Prediction (NTP) way, which can further enhance the efficiency and acquire the controllability.

problem from a complexity-adaptive perspective. We model the control of the injection intensity as the selection of the injection interval, i.e., the number of tokens between two adjacent injections. We propose a complexity-adaptive and dynamic interval control mechanism, formulated as follows:

$$\tau_k = \alpha + \beta \cdot l_k, \ \ \alpha > 0, \ \beta > 0, \tag{1}$$

where $\tau_k$ is the current injection interval. $l_k$ denotes the current length of the reasoning process, i.e., the number of current output tokens, which serves as a complexity indicator herein. $\alpha$ is the basic length of the injection interval, and $\beta$ is a positive coefficient to control the strength of adaptivity. Every time $\tau_k$ is obtained, the model will generate the next $\tau_k$ tokens, inject the hint, and update $l_k$ and $\tau_k$ in sequence. This cycle is repeated until the reasoning process is completed. The injection interval $\tau_k$ is a linear function of the current length, which indicates that the hint interval will increase with the current reasoning length. Here, we hold a prior that the reasoning length of a query is approximately positively correlated with its complexity (Muennighoff et al., 2025; Lee et al., 2025), and the intuitive assumption that easy queries can tolerate a larger compression ratio than complex ones. When the current length $l_k$ is small, the injection interval is set to a small value, resulting in a higher hint intensity. The reasoning of easy queries will complete in a short length, such as hundreds of tokens, so their average hint intensity is high, ensuring a high level of conciseness. If the length continues to increase, it will indicate that this query should be complex rather than easy, so Equation (1) accordingly relieves the hint intensity by increasing the injection interval $\tau_k$, avoiding excessive hinting that harms the accuracy. This adaptive strategy avoids manually setting the injection interval based on precise estimation of the complexity, as it is usually intractable.

**The selection of the hyperparameters $\alpha$ and $\beta$.** $\alpha$ should be set to a small value to ensure conciseness for easy queries, as they can tolerate high injection intensity. Empirical results show the performance is not sensitive to $\beta$, as long as it is not excessively small. Detailed ablation study and discussion about $\alpha$ and $\beta$ can be found in Section A.1. In all our experiments, we fixed $\alpha$ to 128 (a small value) and $\beta$ to 0.2 to avoid manual hyper-parameter tuning, and we find it always works well for various models and benchmarks.

**The dynamic selection strategy for the hint injection position.** Another problem is how to select the position to inject the hint. Let $T$ denote the original generated text whose length is $\tau_k$, $p$ denote the position of injection, and $T_{hint}$ denote the hint. Then, the modified text after hint injection will be:

$$T' = T[0:p] + T_{hint} + T[p:\tau_k - 1], \;\; p \in [0, \tau_k - 1]. \tag{2}$$

We reveal two rules about the selection of injection position $p$: *(i)* $p$ should not be too close to $\tau_k - 1$ to avoid accuracy degradation. Concretely, if $p$ is very close to $\tau_k - 1$, the injected hint will approach the tail of the generated text. In this case, we observe that the subsequent generation will soon terminate the thinking or just lazily repeat the text generated after the last hint (see case studies in Section A.8), which significantly undermines accuracy, as shown in Table 4. *(ii)* $p$ should not be too close to $0$. Although injecting the hint into the head solves the accuracy degradation problem, it introduces extra computing costs caused by prefilling the text between the injection position and the end, i.e., $T[p:\tau_k - 1]$. Therefore, to ensure a good computing-efficiency balance, we propose a dynamic selection strategy for the position $p$, formulated as follows:

$$p = \tau_k * \min(\; (\tau_k - \alpha)/1024, \; 0.8 \;), \tag{3}$$

where $\tau_k$ is the current injection interval and $\alpha$ is the basic injection interval length, the same as those in Equation (1). During the early reasoning, $\tau_k$ is small, so the injection position is close to the head, not suffering from the aforementioned accuracy degradation. As the reasoning proceeds, $\tau_k$ becomes larger, the injection position moves towards the tail to save prefilling costs. Meanwhile, we restrict the maximum position to $\tau_k \cdot 0.8$ to prevent it from being too close to the tail, avoiding the accuracy degradation. The detailed theoretical and empirical analysis for injection costs can be found at Section A.2, which indicates that the extra costs of our strategy are negligible.

---

**Algorithm 1** The proposed ConciseHint algorithm.

---

1: **Input:** input prompt $P$ and model $M$. hint $T_{hint}$, basic interval length $\alpha$, and coefficient $\beta$.
2: $\tau_k = \alpha$, $l_k = 0$. $O_k = P$          ▷ Initialize injection interval, current length, and current output.
3: **while** True **do**
4:      $T$, finish_reason = client.completions.create(model= $M$, prompt= $O_k$, max_token_len= $\tau_k$ ) ▷ Call model generation.
5:      $p = \tau_k * \min(\; (\tau_k - \alpha)/1024, \; 0.8 \;)$          ▷ Compute the injection position.
6:      $T' = T[0:p] + T_{hint} + T[p:\tau_k - 1], \;\; p \in [0, \tau_k - 1].$          ▷ Inject the hint.
7:      $O_k = O_k + T'$          ▷ Update current output.
8:      $l_k = l_k + \tau_k$          ▷ Update current length.
9:      $\tau_k = \alpha + \beta \cdot l_k$          ▷ Update injection inverval.
10:      **if** finish_reason is Stop **then** break
11:      **end if**
12: **end while**
13: Return $O_k$          ▷ Get the overall answer.

---

**ConciseHint-T: training the embeddings of hint on concise reasoning data to learn concise patterns.** Even though the training-free ConciseHint effectively improves the efficiency, further training the hint embeddings can bring additional token reduction. Concretely, firstly, we prepare a dataset consisting of questions and corresponding concise reasoning responses. Next, we construct modified reasoning responses by injecting hint embeddings to be trained into the original responses at a fixed interval. We initialize the hint embeddings as the embeddings of our manually designed hint ($\mathbf{E}_{ori}$) used in ConciseHint. Finally, we conduct supervised fine-tuning (like Prompt Tuning (Lester et al., 2021)) on the questions and corresponding modified responses, following the next-token prediction paradigm, and obtain the optimized hint embeddings $\mathbf{E}_{optim}$. We expect the hint embeddings to learn the inherent concise patterns in the concise reasoning responses. Then, ConciseHint-T uses the optimized hint embeddings and thus further reduces token usage. Moreover, we observe that we can control the token usage through the interpolation between the initial hint embeddings and the optimized embeddings. The interpolation embeddings can be derived from:

$$\mathbf{E}_{interp} = \gamma * \mathbf{E}_{optim} + (1 - \gamma) * \mathbf{E}_{ori}, \ \gamma \in [0, 1] \tag{4}$$

**Controllability can be achieved by adjusting** $\gamma$, where a higher value usually leads to less token usage. $\gamma = 1$ denotes our ConciseHint-T, while $\gamma = 0$ is ConciseHint.

## 4 EXPERIMENTS

### 4.1 EXPERIMENTAL SETUP

**Benchmarks.** Following mainstream practice, we mainly validate our method on three commonly-used benchmarks for large reasoning models, i.e, GSM8K (Cobbe et al., 2021), AIME24 (Committees, 2024), and GPQA-Diamond (Rein et al., 2024). GSM8K(Grade School Math 8K) consists of more than 8,000 high-quality quality linguistically diverse grade school math word problems. We use the test split containing 1,319 problems. AIME24 consists of 30 mathematical problems from the 2024 American Invitational Mathematics Examination (AIME24), a renowned high school math competition recognized for its difficult and thought-provoking problems. GPQA-diamond consists of 198 high-quality and challenging multiple-choice questions written by domain experts in biology, physics, and chemistry. Moreover, in the appendix, we also report results on the commonsense reasoning benchmark CommonsenseQA, and the code generation benchmark HumanEval.

**Models.** We evaluate our method on the state-of-the-art open-source large reasoning models including Qwen3-8B, Qwen3-4B, Qwen3-1.7B (Alibaba, 2025), and DeepSeek-R1-14B (Guo et al., 2025), which deliver remarkable advancements in tackling a wide range of reasoning tasks.

**Baselines.** The basic baseline is the original reasoning without any efficiency technique. Besides, we include four representative efficient methods as baselines. Specifically, BeConcise (Renze & Guven, 2024) is a commonly-used prompting-based method that appends a prompt of "Be concise" to the input to encourage answering concisely. Besides, we obtain a stronger prompting method by adding "Please adaptively control the answer length based on the query's complexity. The lower the complexity, the more concise your answer should be". We denote it as "Prompt" for simplicity. Moreover, we include the early-exit method Deer (Yang et al., 2025), which terminates the reasoning when the model is confident enough. We also include NoWait (Wang et al., 2025), which prohibits transition tokens like "wait" and "alternatively" to obtain more efficient self-reflections.

**Evaluation configurations.** For all experiments, we set the temperature to 0.6 and top-p to 0.95, which is recommended in the official documentation. We report the accuracy to measure model performance. Following mainstream works, we report the average token usage, i.e., the average number of tokens to answer a query, to measure the efficiency. The injected hints are also counted. Each experiment is run multiple times, and we report the average results. For GSM8K, we run 5 times. For others, we run 10 times.

### 4.2 MAIN RESULTS.

**ConciseHint results.** Table 1 shows the main quantitative results of our experiments. Ori. denotes the original reasoning process without any efficiency technique. Ours (baseline) denotes the combination of our ConciseHint and the baseline method. For example, Ours (Ori) means applying ConciseHint in the original reasoning. From the experimental results in Table 1, we can derive the following two key conclusions:

(i) **When individually applied, ConciseHint can effectively improve the reasoning efficiency, which is comparable to strong baselines.** Firstly, compared to the original reasoning (i.e., Ori.), employing ConciseHint (i.e., Ours (Ori)) can effectively reduce the token usage while maintaining the accuracy well. For example, on the GSM8K benchmark and Qwen3-4B, Ours (Ori) reduces 48.9% tokens from 2381 to 1213, with only an accuracy loss of 0.07. On the GPQA Diamond, it reduces 44.5% tokens from 7388 to 4099, even with an accuracy rise of 0.91. Secondly, the efficiency improvement of Ours (Ori) is comparable to these four efficiency baseline methods. For example, on the GSM8K benchmark and Qwen3-4B, the token usage of Ours (Ori) is less than BeConcise (1597), Prompt (1263), Deer (1405) and NoWait (1289). By continuously injecting concise hints, our method effectively reduces the token usage.

Table 1: ConciseHint results on GSM8K, AIME24, and GPQA-Diamond with Qwen3-4B, Qwen3-8B, and Deepseek-R1-14B. Ori. denotes the original reasoning process. Besides, we also include BeConcise (Renze & Guven, 2024), Prompt, Deer (Yang et al., 2025), and NoWait (Wang et al., 2025) as baselines. Ours (baseline) denotes the combination of our ConciseHint and the baseline method. We report the accuracy and average token usage. The lowest token usage is highlighted in bold. The red and blue numbers show the token reduction percentage over the original reasoning and the corresponding baseline method, respectively.

| Model | Method | GSM8K | | AIME24 | | GPQA-Diamond | |
|---|---|---|---|---|---|---|---|
| | | Accuracy% | Token usage | Accuracy% | Token usage | Accuracy% | Token usage |
| Qwen3-4B | Ori. | 94.81 | 2381 | 64.33 | 11634 | 51.82 | 7388 |
| | Ours (Ori) | 94.74 | 1213 (-49%) | 66.67 | 10523 (-10%) | 52.73 | 4099 (-45%) |
| | BeConcise | 94.60 | 1597 | 64.33 | 10929 | 53.74 | 6113 |
| | Prompt | 94.56 | 1263 | 63.67 | 10755 | 52.93 | 5180 |
| | Ours (Prompt) | 94.75 | **839** (-65%/-34%) | 67.00 | 9255 (-20%/-14%) | 51.72 | 3190 (-57%/-38%) |
| | Deer | 94.78 | 1405 | 64.00 | 10149 | 53.23 | 6878 |
| | Ours (Deer) | 94.31 | 841 (-65%/-40%) | 65.33 | **8410** (-28%/-17%) | 52.31 | 3925 (-47%/-43%) |
| | NoWait | 94.33 | 1289 | 59.00 | 10053 | 52.12 | 5246 |
| | Ours (NoWait) | 94.03 | 857 (-64%/-34%) | 58.33 | 8893 (-24%/-12%) | 51.31 | **2730** (-63%/-48%) |
| Qwen3-8B | Ori. | 95.86 | 2382 | 64.67 | 11725 | 57.58 | 8524 |
| | Ours (Ori) | 95.53 | 1489 (-37%) | 67.33 | 11228 (-4%) | 57.68 | 5400 (-37%) |
| | BeConcise | 95.78 | 1822 | 66.67 | 11371 | 57.17 | 7466 |
| | Prompt | 95.72 | 1353 | 68.00 | 10693 | 57.58 | 6285 |
| | Ours (Prompt) | 95.51 | 935 (-61%/-31%) | 69.67 | 9996 (-15%/-7%) | 55.56 | 3880 (-54%/-38%) |
| | Deer | 95.62 | 1223 | 66.33 | 10298 | 55.45 | 7778 |
| | Ours (Deer) | 95.22 | **907** (-62%/-26%) | 64.67 | **8843** (-25%/-14%) | 55.35 | 5306 (-38%/-32%) |
| | NoWait | 95.38 | 1406 | 64.83 | 9936 | 56.67 | 6575 |
| | Ours (NoWait) | 95.06 | 1030 (-57%/-27%) | 64.17 | 9457 (-19%/-5%) | 55.56 | **3860** (-55%/-41%) |
| DeepSeek-R1-14B | Ori. | 95.03 | 981 | 63.00 | 9210 | 56.06 | 5038 |
| | Ours (Ori) | 94.87 | 713 (-27%/-27%) | 61.00 | 7623 (-17%/-17%) | 54.65 | 3715 (-26%/-26%) |
| | BeConcise | 94.92 | 770 | 63.00 | 8521 | 55.96 | 4739 |
| | Ours(BeConcise) | 94.58 | 686 (-30%/-11%) | 62.67 | 7446 (-19%/-13%) | 55.61 | 3883 (-23%/-18%) |
| | Prompt | 94.18 | 627 | 64.67 | 7597 | 55.05 | 4120 |
| | Ours(Prompt) | 94.03 | 590 (-40%/-6%) | 64.67 | 6643 (-28%/-13%) | 54.29 | 3428 (-32%/-17%) |
| | Deer | 94.64 | 663 | 62.67 | 8416 | 54.45 | 4920 |
| | Ours(Deer) | 94.31 | 604 (-38%/-9%) | 62.67 | 7180 (-22%/-15%) | 54.04 | 3915 (-22%/-20%) |

(ii) **When integrated, ConciseHint consistently and obviously enhances the reasoning efficiency across all baseline methods, substantially raising the upper bound of efficiency.** Let us focus on the comparison between Ours (baseline) and the corresponding baseline method. For each baseline method, applying ConciseHint obviously reduces the token usage while maintaining the accuracy well. For example, on the GSM8K benchmark and Qwen3-4B, compared to Deer, Ours (Deer) reduces 40.1% tokens from 1405 to 841. The overall reduction ratio against the original reasoning rises to 65%. Compared to NoWait, Ours (NoWait) reduces 33.5% tokens from 1289 to 857. The overall reduction ratio is 64%. The results validate the flexibility and compatibility of our approach, enabling seamless integration with various existing methods.

Table 2: ConciseHint-T (incorporating training) results on GSM8K, AIME24, and GPQA-Diamond with Qwen3-1.7B. "Ours" and "Ours-T" denote our ConciseHint and ConciseHint-T, respectively. The embeddings are learned on MixChain-Z-GSM8K.

| Method | GSM8K | | AIME24 | | GPQA-Diamond | |
|---|---|---|---|---|---|---|
| | Accuracy | Token usage | Accuracy | Token usage | Accuracy | Token usage |
| Ori. | 90.87 | 2458 | 39.33 | 13570 | 39.39 | 9223 |
| Ours | 90.04 | 1237 | 42.67 | 11859 | 37.37 | 5105 |
| Ours-T ($\gamma = 0.7$) | 90.19 | 996 | 39.00 | 11029 | 37.37 | 4279 |
| Ours-T ($\gamma = 1.0$) | 88.01 | 742 | 40.67 | 10223 | 35.05 | 3776 |

**Incorporating hint training to further enhance the efficiency: ConciseHint-T results.** We train the hint embeddings on the MixChain-Z-GSM8K (Ma et al., 2025) dataset, which consists of concise question-response pairs built on GSM8K training dataset. Table 2 shows the results of ConciseHint-T. At $\gamma = 0.7$, ConciseHint-T achieves additional token reduction over ConciseHint while preserving the accuracy. Increasing $\gamma$ to 1 yields a more substantial reduction, even though at the cost of accuracy degradation on GPQA Diamond. These results indicate that the trained embeddings have effectively captured the concise patterns inherent in the concise reasoning data, thereby enhancing the efficiency

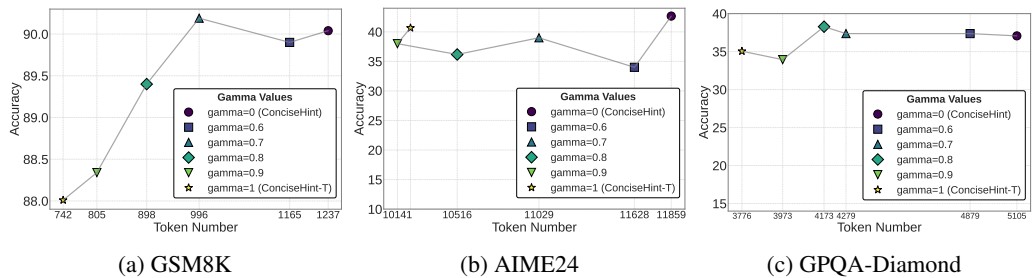

|             |             |                  |
| :---------: | :---------: | :--------------: |
| (a) GSM8K   | (b) AIME24  | (c) GPQA-Diamond |

Figure 3: Controllability curves obtained by adjusting $\gamma$ on Qwen3-1.7B. Different scattered points represent different $\gamma$ values.

over the manually designed hint. Moreover, the results demonstrate that the learned embeddings are not only effective on in-domain data (GSM8K) but also **generalize well to out-of-domain data** (AIME24 and GPQA Diamond).

Figure 3 shows the controllability results by adjusting $\gamma$ in Equation (4). On all datasets, a higher $\gamma$ value always leads to lower token usage. Additionally, it shows that shorter reasoning chains can sometimes achieve higher accuracy, indicating that a longer reasoning chain does not necessarily lead to better performance (Ma et al., 2025).

Table 3: The ablation study on the selection of the injection interval of ConciseHint.

| Model     | Dataset | Inject. interval | Accuracy% | Token usage |
| :-------- | :------ | :--------------- | :-------- | :---------- |
| Qwen3-8B  | AIME24  | Ours (adaptive)  | 69.67     | 9996        |
|           |         | Fixed 64         | 61.67     | 9941        |
|           |         | Fixed 128        | 66.67     | 9757        |
|           | GSM8K   | Ours (adaptive)  | 95.51     | 935         |
|           |         | Fixed 64         | 95.65     | 908         |
|           |         | Fixed 128        | 95.45     | 933         |
| Qwen3-4B  | AIME24  | Ours (adaptive)  | 67.00     | 9255        |
|           |         | Fixed 64         | 45.33     | 6598        |
|           |         | Fixed 128        | 63.33     | 9036        |
|           | GSM8K   | Ours (adaptive)  | 94.75     | 839         |
|           |         | Fixed 64         | 93.42     | 763         |
|           |         | Fixed 128        | 94.44     | 835         |

## 4.3 ABLATION STUDIES

Through ablation studies, we demonstrate the necessity of adaptively controlling the injection intensity based on the complexity (Equation (1)), and the necessity of dynamically determining the position of hint injection (Equation (3)). We also present corresponding cases to make it clearer.

**The necessity of adaptively controlling the injection intensity.** Recall that our method continuously scales up the injection interval to make it positively correlated with the current length. This strategy avoids excessive intervention in complex problems while ensuring a high intensity of intervention in easy problems. We use Table 3 to quantitatively demonstrate it, where "Fixed" denotes that the injection interval is set to the fixed value, and the injection intensity is inversely proportional to the interval. We conduct experiments on AIME24 and GSM8K, as their complexity levels differ a lot. From the results, we can conclude that a high intensity of hint injection impairs the performance of complex queries, but has little effect on simple queries. For example, using the fixed interval of 64 significantly decreases the accuracy of Qwen3-4B from 67.00 to 45.33 on AIME24, but on the GSM8K, the accuracy loss is minor. And it decreases the accuracy of Qwen3-8B from 69.67 to 61.67 on AIME24, but it would even slightly improve the accuracy from 95.51 to 95.65 on GSM8K. Therefore, to avoid performance degradation, it is necessary to relieve the injection intensity for complex queries. In the circumstances where we can know the approximate complexity of a given query in advance, we can just set a larger fixed interval for those complex queries. For example, we

Table 4: The ablation study on the selection of the injection position of ConciseHint. The prefilling ratio denotes the ratio of tokens to be prefilled after hint injection.

| Model | Dataset | Inject. postion | Accuracy% | Token usage | Prefilling ratio% |
|---|---|---|---|---|---|
| Qwen3-8B | GPQA-Diamond | Our Dynamic | 55.56 | 3880 | 0.0 to 0.8 (dynamic) |
| | | At the tail | 42.93 | 1321 | 0.0 |
| | | In the middle | 55.05 | 4443 | 0.5 |
| | | At the head | 58.95 | 3798 | 1.0 |

know the AIME24 is a challenging benchmark. However, it is intractable to precisely measure the complexity of a wild query, and we do not want to turn it into a hyper-parameter selection problem. Therefore, adaptively adjusting the interval using our Equation (1) is essential, as it can automatically adapt to different levels of complexity.

**The necessity of dynamically determining the position of hint injection.** We discuss the influence of the selection of the injection position. Recall that as the reasoning proceeds, our method dynamically moves the injection position from the head towards the tail, to avoid accuracy degradation and save computing. We compare our method to three fixed position selection strategies, i.e., injecting at the tail, in the middle, and at the head. The experimental results in Table 4 indicate that the closer the fixed position is to the head, the better the accuracy it achieves. Specifically, injecting at the tail induces a significant accuracy degradation, from 55.25 to 43.03. Injecting in the middle achieves a comparable accuracy to ours, but causes the rise of token usage. Moreover, although injecting at the head slightly improves the accuracy, it increases the computing a lot due to the 100% token prefilling. Therefore, to avoid both accuracy degradation and computing increase, our dynamic position selection is essential. Section A.2 elaborates on the analysis of prefilling costs, and shows the extra costs of our method are negligible.

## 4.4 THE STATISTICS OF TRANSITION WORDS WHEN SPEAKING CONCISELY

The appearance of transition words (i.e., "Wait" and "Alternatively") often marks the beginning of a new thought step for self-reflection. To investigate the impact on the self-reflection, we compare the average number of transition words and the average interval length between two words, presented in Table 5. It indicates that our method reduces a large proportion of redundant transition words (i.e., redundant thought steps), thereby promoting efficient self-reflections and making the overall reasoning more concise.

Table 5: The statistics of transition words.

| Model | Method | GSM8K | | | GPQA-Diamond | | |
|---|---|---|---|---|---|---|---|
| | | # Token | # Transition words | Transition interval | # Token | # Transition words | Transition interval |
| Qwen3-4B | Ori. | 2381 | 14.97 | 113.42 | 7388 | 59.92 | 102.05 |
| | Ours (Ori) | 1213 | 4.39 | 118.66 | 4099 | 32.08 | 95.55 |
| Qwen3-8B | Ori. | 2382 | 14.05 | 115.77 | 8524 | 66.36 | 105.38 |
| | Ours (Ori) | 1489 | 5.50 | 126.91 | 5400 | 38.17 | 107.92 |

## 5 CONCLUSION

We propose an in-reasoning intervention framework dubbed ConciseHint to boost the efficient reasoning of large reasoning models. Different from mainstream methods that try to enhance the efficiency before the actual reasoning, we highlight a promising paradigm of performing intervention during the generation of the reasoning to make it more concise. ConciseHint injects learnable hints (manually designed or learned on the concise data) into the reasoning process to encourage conciseness. To avoid accuracy degradation for complex queries due to excessive hints, ConciseHint adaptively controls the injection intensity according to the complexity of the query. Besides, it dynamically adjusts the injection position to achieve a good computing-accuracy balance. We conduct experiments on GSM8K, AIME24, and GPQA-Diamond benchmarks with the state-of-the-art reasoning models DeepSeek-R1 and Qwen3 series. The results demonstrate that ConciseHint effectively improves the reasoning efficiency while maintaining the performance well, indicating that

the in-reasoning intervention is a promising direction for boosting reasoning efficiency. Moreover, the results demonstrate that ConciseHint can serve as a flexible plugin that seamlessly integrates with existing methods to further enhance efficiency.

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

# A APPENDIX

## A.1 ABLATION STUDY AND ANALYSIS OF HYPER-PARAMETERS

In the main experiments, we set $\alpha$ and $\beta$ to 128 and 0.2, respectively, and find that it works well across different benchmarks and models. Here, we demonstrate the principles behind this choice and systematically investigate the influence of different values of our hyper-parameters $\alpha$ and $\beta$. Figure 4 shows the ablation results of $\beta$, from which we can obtain the following observations:

- When $\beta$ is greater than 0.2, the improvement in accuracy brought by further increasing beta is not significant.
- When $\beta$ is less than 0.1 (especially equal to 0), it will cause an obvious accuracy degradation on difficult benchmarks (AIME24 and GPQA-Diamond).

It indicates the performance will not be sensitive to $\beta$, as long as it is not excessively small. Therefore, $\beta$ should be greater than a certain threshold, e.g., $\geq 0.2$ in our settings. So, we set $\beta$ to 0.2 in our main experiments.

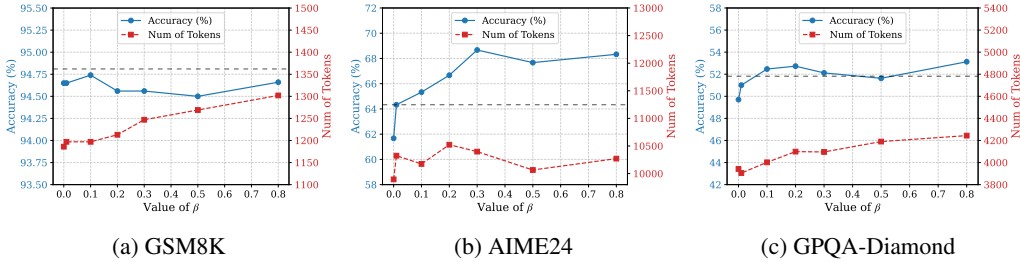

| (a) GSM8K | (b) AIME24 | (c) GPQA-Diamond |

Figure 4: Ablation study of $\beta$ on GSM8K, AIME24, and GPQA-Diamond with Qwen3-4B. The black line denotes the accuracy of the original reasoning.

Table 6 shows the results of different $\alpha$, from which we observe:

- A larger $\alpha$ generally induces a greater number of output tokens and a better accuracy.
- The impact of $\alpha$ on accuracy is more significant on difficult benchmarks (GPQA-Diamond) than on simple benchmarks (GSM8K).

Therefore, to ensure conciseness on easy benchmarks, $\alpha$ should be set to a small value (e.g., 128 and 256). Besides, $\alpha$ should not be excessively small (e.g., 64 or less), aiming to obtain a good accuracy-efficiency balance for difficult benchmarks.

Table 6: Ablation study of $\alpha$ on GSM8K and GPQA-Diamond with Qwen3-4B.

| $\alpha$ | GSM8K | | GPQA-Diamond | |
|---|---|---|---|---|
| | Acc (%) | # Tokens | Acc (%) | #Tokens |
| 64 | 94.78 | 1062 | 51.92 | 3920 |
| 128 | 94.74 | 1213 | 52.73 | 4099 |
| 256 | 94.74 | 1370 | 53.54 | 4347 |
| 512 | 94.81 | 1571 | 53.43 | 4675 |
| 1024 | 94.92 | 1851 | 53.74 | 4986 |

## A.2 THEORETICAL AND EMPIRICAL ANALYSIS FOR INJECTION COSTS

The extra costs of injecting hint derive from the prefilling of tokens after the injection position. We use Figure 5 to visualize this process. According to our ConciseHint algorithm (Algorithm 1), in each iteration, we first generate the text $T$ and meanwhile cache tokens in $T$ (marked in green). Then, we inject the hint into $T$, which makes the cached KV values after the injection position invalid

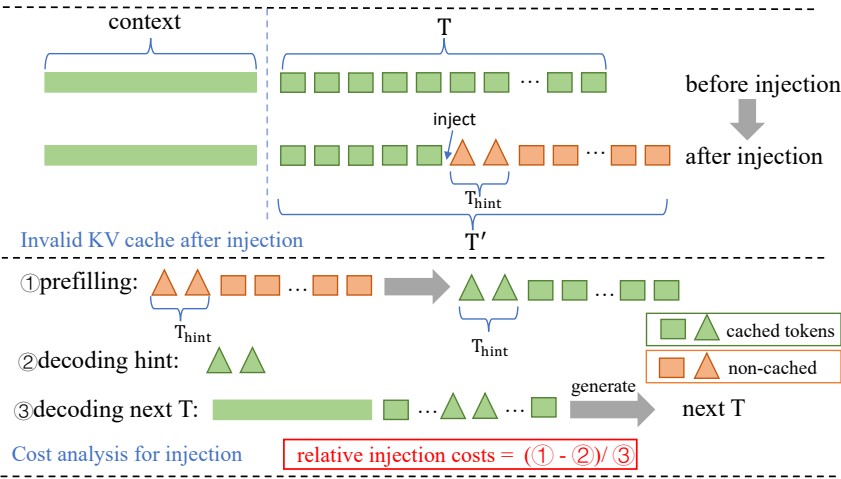

Figure 5: Visualization of cost analysis for hint injection. Green and orange represent cached and non-cached tokens, respectively. The rectangles and triangles represent common and injected hint tokens, respectively.

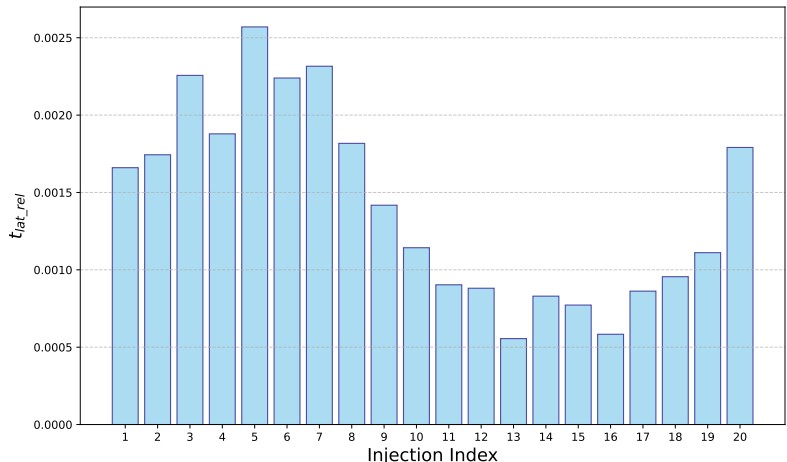

Figure 6: Empirical evaluation of the relative latency on AIME24 with Qwen3-4B. The x-axis represents the injection index, and the y-axis represents the relative latency caused by this hint injection. Injection index=N means that we have previously injected N-1 hints in this reasoning, and the current one is the N-th hint.

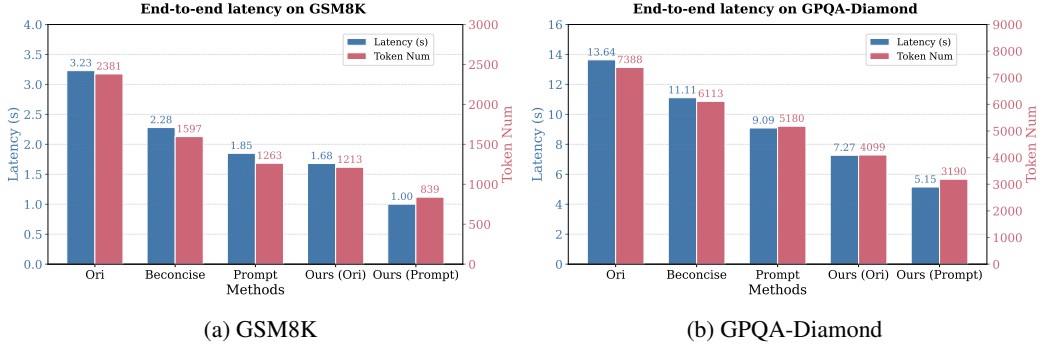

(a) GSM8K

(b) GPQA-Diamond

Figure 7: End-to-end latency of Qwen3-4B on GSM8K and GPQA-Diamond. The vLLM (Kwon et al., 2023) library is employed to perform inference. The data is collected on NVIDIA RTX 6000 with a batch size of 64.

(marked in orange). So, when generating the next token, these KV values need to be recomputed, like a prefilling stage. Let $t_{pre}$ denote the latency of the prefilling stage. Based on Algorithm 1, the number of tokens needing prefilling is $\tau_k - p$ (for simplicity, we ignore the length of the hint, as it is short). On the other hand, we also save time due to injecting tokens, as these tokens appear immediately without token-by-token decoding, and they also account for the total number of tokens. Let $t_{dec\_hint}$ denote the saving time. Therefore, the equivalent absolute latency caused by each hint injection is: $t_{lat\_abs} = t_{pre} - t_{dec\_hint}$. As is well known, the prefilling is usually much faster than token-by-token decoding. So, the values of $t_{lat\_abs}$ are usually low. Besides, a more meaningful metric is the relative latency, i.e., $t_{lat\_rel} = \frac{t_{lat\_abs}}{t_{dec\_text}}$, where $t_{dec\_text}$ is the latency of generating the text $T$ in the subsequent iteration. $t_{lat\_rel}$ measures the proportion of extra latency w.r.t the original generation latency. The relative latency is low, which is also because the decoding is more costly.

Empirical evaluation of the relative latency $t_{lat\_rel}$ is shown in Figure 6, which indicates the extra costs of our ConciseHint are negligible (less than 0.3%).

Empirical **end-to-end latency** evaluations are presented in Figure 7, which demonstrates that our method significantly reduces the actual reasoning latency. For example, on GSM8K, Ours (Ori) reduces the latency from 3.23s to 1.68s, and Ours (Prompt) further reduces it to only 1.00s.

## A.3 EVALUATION ON CODE GENERATION AND COMMONSENSE REASONING.

To further validate the broad applicability of our approach, we conduct additional experiments on code generation and commonsense reasoning tasks. The results of Qwen3-4B are shown Table 7. These results show the efficacy of our method to facilitate efficient reasoning on code generation and commonsense reasoning.

Table 7: The results on the code generation benchmark HumanEval (Chen, 2021) and the commonsense reasoning benchmark CommonsenseQA (Talmor et al., 2019).

| Method | HumanEval | | | CommonsenseQA | |
|---|---|---|---|---|---|
| | pass@1 | pass@10 | Token usage | Accuracy | Token usage |
| Ori. | 93.96 | 98.78 | 2715 | 81.00 | 657 |
| Ours (Ori) | 94.20 | 96.34 | 2355 | 80.81 | 463 |
| Prompt | 93.35 | 96.95 | 2688 | 80.59 | 485 |
| Ours (Prompt) | 93.07 | 97.56 | 1983 | 80.30 | 408 |
| Deer | 93.41 | 96.95 | 2685 | 80.11 | 410 |
| Ours (Deer) | 93.12 | 96.34 | 2218 | 80.00 | 378 |

## A.4 IMPACT ON REASONING CLARITY OF OUR HINT

To directly assess whether hints change reasoning clarity, we conducted an additional pairwise evaluation using GPT-4o-mini as an external judge. Given two reasoning chains (with and without hints) for the same question, the judge was asked: "Given two reasoning chains, A and B, which one has higher quality, i.e., is more logically coherent, consistent, and structurally sound? Or are they equally good?" For correct responses, the judge labels the two chains as equal quality in over 99.5% of cases. This strongly suggests that ConciseHint does not degrade clarity, logical consistency, or structural soundness.

## A.5 SENSITIVITY STUDY ON DIFFERENT HINT DESIGN

We conduct a sensitivity study on different hint designs, whose results are shown in Table 8. Different hint designs consistently reduce the token usage and maintain the accuracy well.

## A.6 COMPARISON TO ADDITIONAL BASELINES

We report comparison results to additional baselines AlphaOne (Zhang et al., 2025b) and O1-pruner (Luo et al., 2025). The results on DeepSeek-R1-Distill-Qwen-7B are shown in Table 9 and Table 10, respectively. Across all benchmarks, our method consistently outperforms these baselines.

Table 8: Sensitivity study on different hint design.

| Hint design | GSM8K | | AIME24 | | GPQA-Diamond | |
|---|---|---|---|---|---|---|
| | Accuracy | Token usage | Accuracy | Token usage | Accuracy | Token usage |
| - (baseline) | 94.81 | 2381 | 64.33 | 11634 | 51.82 | 7399 |
| make answer concise! | 94.74 | 1213 | 66.67 | 10523 | 52.73 | 4099 |
| Be succinct, no redundancy! | 94.89 | 1296 | 66.67 | 10785 | 52.43 | 4443 |
| Keep response brief and tight! | 94.71 | 1092 | 66.67 | 9311 | 52.53 | 3678 |
| Prioritize brevity, avoid verbosity! | 95.07 | 1366 | 65.00 | 10867 | 54.33 | 4613 |

Table 9: Comparison to AlphaOne.

| Method | AIME24 | | AMC23 | | MATH-500 | |
|---|---|---|---|---|---|---|
| | Accuracy | Token usage | Accuracy | Token usage | Accuracy | Token usage |
| Ori. | 46.7 | 6648 | 82.5 | 4624 | 87.6 | 3239 |
| AlphaOne | 50.0 | 6827 | 90.0 | 4397 | 91.2 | 4337 |
| Ours (Ori) | 50.0 | 6155 | 90.0 | 3953 | 91.0 | 2630 |

## A.7 EVALUATION ON LARGER LRM

We validate our method on a larger LRM Qwen3-30B-A3B (Alibaba, 2025), the results on GSM8K are shown in Table 11, demonstrating the efficacy of our method on larger LRMs.

## A.8 CASE STUDIES

We present case studies in Figure 8 to demonstrate the ablation studies (Section 4.3) more clearly, showing how the injection intensity and position affect the reasoning. The upper panel shows a sample from AIME24 benchmark with Qwen3-4B when the injection interval is fixed to 64, leading to a high hint intensity (corresponds to row 9 in Table 3). We can see that the model directly terminates the output after generating "Rotation by 135° (3/8 of a full rotation)" and gives no final answer under the intensive hint, resulting in a significant accuracy degradation. The lower panel shows a sample from GPQA-Diamond benchmark with Qwen3-8B when the hint is injected at the tail (corresponds to row 3 in Table 4). On the first query, the model suddenly ends the thinking after "Let me recall: benzoquinone has two carbonyl groups", and then gives its final answer, and this sort of insufficient thinking will reduce the accuracy. On the second query, the model repeats the text between two adjacent hints, i.e., "Step-by-step explanation: step1... step2... step3". The underlying reason lies in the fact that generating tokens immediately following the hint predisposes the model to lazily recycle textual outputs previously generated following the last hint, which will harm the performance. Therefore, moving the injection position forward can alleviate this problem, which leaves enough tokens after the hint to make the subsequent generation stable.

## A.9 THE USE OF LARGE LANGUAGE MODELS

In this paper, including the main body and appendix, large language models are used solely to polish writing, and only for a small portion of sentences.

Table 10: Comparison to O1-Pruner.

| Method | AIME24 | | GSM8K | | MATH-500 | |
|---|---|---|---|---|---|---|
| | Accuracy | Token usage | Accuracy | Token usage | Accuracy | Token usage |
| Ori. | 53.5 | 10306 | 87.9 | 682 | 90.2 | 3674 |
| O1-Pruner | 49.2 | 9719 | 87.6 | 428 | 86.6 | 2534 |
| Ours (Ori) | 57.7 | 8853 | 88.0 | 400 | 90.8 | 2658 |

Table 11: Evaluation results on Qwen3-30B-A3B

| Method | GSM8K | |
|---|---|---|
| | Accuracy | Token usage |
| Ori. | 96.38 | 1272 |
| Ours (Ori) | 96.47 | 999 |
| Prompt | 95.83 | 785 |
| Ours (Prompt) | 96.74 | 763 |
| Deer | 96.13 | 902 |
| Ours (Deer) | 96.03 | 686 |
| Nowait | 95.91 | 1086 |
| Nowait (Ours) | 96.21 | 879 |

**Intensive hint for complex query**

**Q:** Each vertex of a regular octagon is independently colored either red or blue with equal probability. The probability that the octagon can then be rotated so that all of the blue vertices end up at positions where there were originally red vertices is $\frac{m}{n}$ where m and n are relatively prime positive integers. What is m+n?

**Reasoning:** [hint] Okay, so I need to find the probability that, after coloring each vertex of a regular octagon either red or blue… Then, express this probability as a reduced fraction m/n and find m + n [hint] … Wait, [hint] for example, if we have a rotation r with orbit size k, then for each vertex in the orbit, if it is blue… [hint] The cycle structure is 2 cycles of length 4. Therefore, the number of colorings is (F_{3} + F_{5})^2 = (2 + 5)^2 = 7^2 = 49. Rotation by 135° (3/8 of a full rotation) <directly end the output>

**Position at the tail**

**Q:** Compounds that have the same molecular formula but are different in their structural arrangement are known as isomers… What is the correct option to this question?
**Reasoning:** Okay, let's tackle this question step by step… Let me recall: benzoquinone has two carbonyl groups [hint] <end thinking and give the final answer> The correct answer is C.

**Q:** How many of the following compounds will exhibit optical activity?… What is the correct option to this question?
**Reasoning:** Okay, let me try to figure out how many of these compounds exhibit optical activity… [hint] Step-by-step explanation: step1… step2… step3… [hint] Step-by-step explanation: step1… step2… step3… [hint] Step-by-step explanation: step1… step2… step3… <repeat the text between adjacent hints> The correct answer is D.

Figure 8: Case studies for using intensive hints for complex queries and injecting the hint at the tail of the original text. Samples from AIME24 and GPQA-Diamond. The orange bar represents the injected hints.

