# OpenReview forum: "ConciseHint: Boosting Efficient Reasoning via Continuous Concise Hints during Generation"
_ICLR.cc/2026/Conference — Submitted to ICLR 2026_

### Official Review · Reviewer_PxsQ · 2025-10-28

**Soundness:** 2
**Presentation:** 2
**Contribution:** 3
**Rating:** 4
**Confidence:** 4

**Summary:**

This paper proposes ConciseHint, a framework that improves reasoning efficiency in large reasoning models by injecting concise hints during generation rather than before reasoning or through fine-tuning. The method continuously introduces either manually designed or learned hints to encourage concise thinking while maintaining accuracy. It adaptively controls the hint intensity and injection position based on query complexity through parameters α and β. The extended version, ConciseHint-T, learns hint embeddings from concise reasoning data and introduces a controllable interpolation parameter γ. Experiments on GSM8K, AIME24, and GPQA-Diamond using Qwen and DeepSeek-R1 show token reduction with minimal accuracy loss.

**Strengths:**

- The proposed training-free ConciseHint framework introduces an interesting perspective by applying in-reasoning intervention rather than pre-reasoning prompting or fine-tuning.
- The paper addresses an important and timely question about improving reasoning efficiency in large reasoning models.
- The experimental results are promising, showing substantial token reduction while maintaining accuracy across the Qwen family of models.
- The idea of adaptive hint injection and its potential for plug-and-play integration with existing efficiency techniques is conceptually appealing.

**Weaknesses:**

1. Hyperparameter Selection and Clarity
- The strategy for determining hint injection intensity (parameters α and β) appears ad hoc.
- Although the paper claims that hints are “learnable,” the positions and frequencies of injection are determined through manually tuned hyperparameters rather than learned mechanisms.
- The description of α and β is confusing and inconsistent. For example, the paper suggests α should be “small,” yet sets α = 128 without clarifying what range is considered small or providing sufficient justification through sensitivity analysis

2. Overreliance on Hyperparameters
- The effectiveness of ConciseHint heavily depends on α, β, and γ, which undermines its claim of adaptivity.
- The framework’s practicality is limited without a clear or automated method for selecting these hyperparameters. This raises concerns about reproducibility and robustness across unseen models. Obviously, the current hyperparameter settings are good for Qwen family models, but not as significantly effective in DeepSeek models.

3. Incomplete Experimental Coverage
- The comparison with baselines is uneven across models. For instance, Qwen3-4B includes BeConcise and other baselines, but DeepSeek-R1-14B results omit some of these.
- It is unclear why BeConcise or similar prompting-based baselines cannot be combined with ConciseHint for stronger comparisons. These inconsistencies suggest the experiments are not yet fully comprehensive or standardized.

4. Overclaim on Reasoning
- The evaluation focuses primarily on math and physics QA datasets (GSM8K, AIME24, GPQA-Diamond). Such domains do not represent general reasoning; broader evaluations on coding, commonsense, or multimodal reasoning datasets are needed.
- As a result, the paper’s claim of improving “reasoning efficiency” in general is somewhat overstated.

5. Trade-off Between Accuracy and Efficiency
- In Table 2, performance noticeably degrades when γ increases (e.g., γ = 1), indicating potential instability or limited generalization.
- The results reveal a clear trade-off between conciseness and accuracy, which should be analyzed more thoroughly rather than only reported. A discussion of this trade-off would help readers better understand practical deployment choices.

6. Writing and Presentation Issues
- The paper is difficult to read, with dense notation and numerous hyperparameters that could be summarized more clearly.
- Several equations (e.g., Eq. (1)–(3)) could benefit from intuitive explanations or visual aids describing the adaptive behavior.

**Questions:**

Please refer to the weaknesses.

---

> ### Author Response · Authors · 2025-11-24
> **Response to Reviewer  #PxsQ, Part1**
>
> **Thank you for the constructive comments and suggestions. We respond to your concerns point by point as follows, and are glad to discuss with you.**
>
> ### **W1 & 2. Hyperparameter Selection and Clarity; Overreliance on Hyperparameters**
>
> Sorry for the confusion, and **we’d like to clarify we have conducted extensive hyperparameter sensitivity experiments and provide clear selection guidelines for users. Please refer to Section A.1.**  Specifically, the main observations for the training-free ConciseHint are:
>
> - **Effect of α.** A larger α increases the number of output tokens, typically improving accuracy; Excessively small α (≤64) harms the accuracy; The impact of α on accuracy is more significant on difficult benchmarks (GPQA-Diamond)
> than on simple benchmarks (GSM8K)
> - **Effect of β.** When β < 0.1 (especially β = 0), the accuracy drops noticeably; When β is greater than 0.2, the improvement in accuracy brought by further increasing beta
> is not significant.
>
> **Hence, we provide the guidelines:**
>
> - L661: we recommend **β ≥ 0.2**  for stable performance
> - L683: We therefore recommend **α to be set to a small but not excessively small value, e.g, 128, 256** for a good efficiency-accuracy balance (L683).  We will make it clear to avoid confusion.
>
> Besides, for the trained version ConciseHint-T:
>
> - **Effect of γ.**  γ is used to control the token-accuracy tradeoff. As shown in **Figure 3**, a larger γ results in more aggressive compression (fewer tokens) but may lead to a larger accuracy drop. γ exposes a *user-controllable knob* that allows practitioners to select the desired compression ratio depending on deployment constraints. The trained version aims to aggressively reduce the token usage. When accuracy is the primary concern, users can simply use the training-free ConciseHint.
>
> We believe these analyses and guidelines can help users choose proper hyperparameters.
>
> ### **W3.  Incomplete Experimental Coverage**
>
> - **(a). Additional experiments on DeepSeek-R1-14B.** Following the reviewer’s suggestion, we conducted the missing baseline comparisons for DeepSeek-R1-14B. The results are shown in the following table. These additions confirm that our method consistently improves efficiency across all baselines for **DeepSeek-R1-14B,** strengthening the overall experimental completeness.
>
> |  | GSM8K |  | AIME24 |  | GPQA-D |  |
> | --- | --- | --- | --- | --- | --- | --- |
> | Method | Acc. | #Token | Acc. | #Token | Acc. | #Token |
> | Ori. | 95.03 | 981 | 63.00  | 9210 | 56.06  | 5038 |
> | Ours (Ori) | 94.87 | 713 | 61.00 | 7623 | 54.65 | 3715 |
> |  |  |  |  |  |  |  |
> | BeConcise | 94.92 | 770 | 63.00 | 8521 | 53.20 | 4709 |
> | Ours (BeConcise) | 94.58 | 686 | 62.67 | 7446 | 52.86 | 3883 |
> |  |  |  |  |  |  |  |
> | Prompt  | 94.18 | 627 | 64.67 | 7597 | 55.05 | 4120 |
> | Ours (Prompt) | 94.03 | 590 | 64.67 | 6643 | 54.29 | 3428 |
> |  |  |  |  |  |  |  |
> | Deer | 94.63 | 663 | 62.67 | 8416 | 54.55 | 4920 |
> | Ours (Deer) | 94.31 | 604 | 62.67 | 7180 | 54.04 | 3915 |
> |  |  |  |  |  |  |  |
> | Nowait | 94.69 | 782 | - | - | 52.73 | 3369 |
> | Nowait (Ours) | 94.13 | 670 | - | - | 52.22 | 2534 |
>
> where Nowait fails on AIME24 (causes significant acc loss), so we skip its results.
>
> - **(b). On combining prompting-based baselines with ConciseHint.** In fact, we already evaluated such combinations in the original submission: Table 1 includes *Prompt + Ours*, where **Prompt** refers to a stronger prompting-based strategy compared to BeConcise (as noted in L293). Since Prompt outperforms BeConcise, we initially focused on testing Prompt + Ours to demonstrate the upper bound of synergy between prompting and our hint-based compression. Moreover, the Table in (a) also shows the results of **BeConcise + Ours,** demonstrating the effectiveness
>
> ### **W4.  Overclaim on Reasoning**
>
> Following the reviewer’s suggestion, we conduct additional experiments on **code generation** and **commonsense reasoning** tasks to further validate the broad applicability of our approach. The results of Qwen3-4B are shown in the following tables:
>
> | Dataset | Method | pass@1 | pass@10 | #Token |
> | --- | --- | --- | --- | --- |
> | **Code generation-Human Eval [1]** | Ori. | 93.96 | 98.78 | 2715 |
> |  | Ours (Ori) | 94.20 | 96.34 | 2355  |
> |  | Prompt | 93.35 | 96.95 | 2688 |
> |  | Ours (Prompt) | 93.07 | 97.56 | 1983  |
> |  | Deer | 93.41 | 96.95 | 2685 |
> |  | Ours (Deer) | 93.12 | 96.34 | 2218 |
>
> | Commonsense reasoning-CSQA[2] | Method | Acc | #Token |
> | --- | --- | --- | --- |
> |  | Ori. | 81.00 | 657 |
> |  | Ours (Ori) | 80.81 | 463 |
> |  | Prompt | 80.59 | 485 |
> |  | Ours (Prompt) | 80.30 | 408 |
> |  | Deer | 80.11 | 410 |
> |  | Ours (Deer) | 80.00 | 378 |
>
> These results show the efficacy of our method to facilitate efficient reasoning on code generation and commonsense reasoning. We hope these results indicate the broad applicability of our approach in different reasoning tasks.

---

> ### Author Response · Authors · 2025-11-24
> **Response to Reviewer #PxsQ, Part2**
>
> ### **W5. Discussion about the Trade-off Between Accuracy and Efficiency**
>
> - For the training-free version ConciseHint, the accuracy-efficiency trade-off is controlled by the hyperparameters **α and β. Please refer to the response to W2 for the effects of α and β on the trade-off and corresponding selection guidelines.**
> - For the trained version ConciseHint-T,  γ is used to control the accuracy-efficiency trade-off. As shown in **Figure 3**, a larger γ results in more aggressive compression (fewer tokens) but may lead to a larger accuracy drop. γ exposes a *user-controllable knob* that allows practitioners to select the desired compression ratio depending on deployment constraints. The trained version aims to aggressively reduce the token usage. When accuracy is the primary concern, users can use a lower γ or the training-free ConciseHint.
>
> We think the accuracy-efficiency trade-off is inherent in LRMs. It is expected that extreme compression will lead to a decline in performance.
>
> ### **W6. Writing and Presentation Issues**
>
> Thank you for your suggestion. We use Figure 5 to visualize the hint injection process. We use Section A.1 to summarize the hyperparameters. We use Algorithm 1 to show the pseudo-code of our pipeline. We hope they can help readers understand our method better.
>
> **Thank you for your reading. We hope the additional experiments could enhance our paper.**
>
> Reference:
>
> [1] Evaluating Large Language Models Trained on Code
>
> [2] CommonsenseQA: A Question Answering Challenge Targeting Commonsense Knowledge

---

### Official Review · Reviewer_Hd4V · 2025-10-31

**Soundness:** 3
**Presentation:** 2
**Contribution:** 2
**Rating:** 6
**Confidence:** 4

**Summary:**

To address the inefficiency of Large Reasoning Models (LRMs) caused by verbose Chain-of-Thought (CoT) generation (e.g., redundant tokens and self-checks), this paper proposes ConciseHint, an in-reasoning intervention framework. Unlike existing methods that only intervene before reasoning (e.g., input prompting, SFT/RL), ConciseHint continuously injects learnable hints (manually designed text or data-learned embeddings) during the reasoning generation process. It adaptively adjusts hint intensity based on reasoning length to avoid over-intervention on complex queries, and dynamically shifts injection positions (from head to tail, capped at \(\tau_{k} \cdot 0.8\)) to balance accuracy and computational cost. An enhanced variant, ConciseHint-T, further optimizes hints via supervised fine-tuning on concise data, enabling controllable reasoning length through embedding interpolation. Experiments on GSM8K, AIME24, and GPQA-Diamond with models like Qwen3 series and DeepSeek-R1 show that ConciseHint reduces tokens by 27%-65% while maintaining accuracy, and seamlessly integrates with existing methods (e.g., BeConcise, Deer) to further boost efficiency.

**Strengths:**

Novel In-Reasoning Intervention Paradigm: Breaks the limitation of "pre-reasoning intervention" in existing works, directly guiding conciseness during token generation—opening a new direction for efficient LRMs.

Adaptive and Dynamic Mechanisms: Designs complexity-aware hint intensity (adapting to query difficulty via reasoning length) and dynamic injection positions, ensuring accuracy while maximizing efficiency.

Flexible and Controllable Hint Design: Supports both training-free manual hints and data-learned hints (ConciseHint-T), with interpolation-based controllability to balance token reduction and performance.

Strong Compatibility: Serves as a plug-and-play plugin that integrates seamlessly with existing efficient methods, pushing the upper bound of reasoning efficiency without modifying the base model.

**Weaknesses:**

The largest model tested is 14B (DeepSeek-R1-14B)—no validation on ultra-large LRMs (70B+, e.g., Qwen3-72B, GPT-4o) where CoT verbosity and computational costs are more severe. Larger models often have more stable reasoning chains; it is unclear if ConciseHint’s intervention is redundant or still effective here.

Lack of Redundancy Targeting and Parameter SensitivityWeakness Details:Unquantified Redundancy Suppression: The paper claims ConciseHint reduces "redundant tokens and self-checks" but provides no breakdown of which specific redundancies are eliminated (e.g., transition words like "wait", repetitive premise restatements, logically superfluous steps). Without this analysis, readers cannot confirm if the framework targets meaningful redundancy (vs. accidental suppression of critical logic).Unvalidated Adaptive Parameters: The default values for \(\alpha=128\) (base interval) and \(\beta=0.2\) (adaptive coefficient) are provided without parameter sensitivity analysis. It is unknown how these values perform across tasks (e.g., AIME24’s long reasoning vs. GSM8K’s short steps) or if optimal parameters exist for different scenarios—hindering practical adoption.Hint Content Impact Ignored: The paper tests only one manual hint ("make answer concise!") and a single training dataset (MixChain-Z-GSM8K) for ConciseHint-T.

Lack of Error Analysis and Edge Case RobustnessWeakness Details:Unanalyzed Accuracy Drops: The paper notes minor accuracy drops (e.g., Qwen3-1.7B on GSM8K: 90.87% → 88.01% for \(\gamma=1.0\)) but provides no analysis of why these drops occur.

**Questions:**

see Weaknesses

---

> ### Author Response · Authors · 2025-11-24
> **Response to Reviewer #Hd4V**
>
> **Thank you for the constructive comments and suggestions. We respond to your concerns point by point as follows  and are glad to discuss with you.**
>
> ### **W1. Test on larger model**
>
> In response, we conducted additional experiments on **larger LRMs**. Since Qwen3 does not offer a 72B model and the 235B version exceeds our computational budget, we evaluated ConciseHint on **Qwen3-32B.**  The results are shown in the following table. Our method continues to **substantially reduce reasoning length on various baselines while preserving accuracy**. This indicates that the method is **not redundant** for larger models; even though their reasoning is more stable, ConciseHint still provides significant efficiency gains.
>
> | Dataset | Method | Acc | #Token |
> | --- | --- | --- | --- |
> | GSM8K | Ori.  | 96.38 | 1272 |
> |  | Ours (Ori) | 96.47 | 999 |
> |  | Prompt  | 95.83 | 785 |
> |  | Ours (Prompt) | 96.74 | 763 |
> |  | Deer | 96.13 | 902 |
> |  | Ours (Deer) | 96.03 | 686 |
> |  | Nowait | 95.91 | 1086 |
> |  | Nowait (Ours) | 96.21 | 879 |
>
> ### W2. Redundancy Targeting,  Parameter Sensitivity Analysis, and Hint Content Impact
>
> - **Redundancy Targeting.** Section 4.4 demonstrate which specific redundancies are eliminated by our method. Specifically,  Table 5 shows that our method reduces a large proportion of redundant transition words like “wait” and “alternatively”, thereby promoting efficient self-reflections and making the overall reasoning more concise.
> - **Parameter Sensitivity Analysis.**  We have conducted extensive hyperparameter sensitivity studies and provided concrete **hyperparameter selection guidelines in Section A.1**.  Specifically,
>     - **Effect of α.**  A larger α increases the number of output tokens, typically improving accuracy.   Excessively small α (≤64) harms the accuracy. We therefore recommend **α to be set a small but not excessive small value, e.g, 128, 256** for good efficiency-accuracy balance  (L683).
>     - **Effect of β.** When β < 0.1 (especially β = 0), the accuracy drops noticeably. Hence, we recommend **β ≥ 0.2** (L661) for stable performance.
> - **Hint Content Impact.** Following the reviewer’s suggestion, we additionally conduct a **study on the impact of different hint designs**, as shown in the following table. Different hint designs consistently reduce the token usage and maintain the accuracy well.
>
> | Dataset | Method | Acc | #Token | Hint design |
> | --- | --- | --- | --- | --- |
> | GSM8K | Ori. | 94.81 | 2381 | - |
> |  | Ours (Ori) | 94.74 | 1213 | make answer concise! |
> |  | Ours (Ori) | 94.89 | 1296 | Be succinct, no redundancy!  |
> |  | Ours (Ori) | 94.71 | 1092 | Keep response brief and tight! |
> |  | Ours (Ori) | 95.07 | 1366 | Prioritize brevity, avoid verbosity! |
> |  |  |  |  |  |
> | AIME24 | Ori. | 64.33  | 11634 | - |
> |  | Ours (Ori) | 66.67  | 10523 | make answer concise! |
> |  | Ours (Ori) | 66.67 | 10785 | Be succinct, no redundancy!  |
> |  | Ours (Ori) | 66.67 | 9311 | Keep response brief and tight! |
> |  | Ours (Ori) | 65.00 | 10867 | Prioritize brevity, avoid verbosity! |
> |  |  |  |  |  |
> | GPQA | Ori. | 51.82 | 7399 | - |
> |  | Ours (Ori) | 52.73 | 4099 | make answer concise! |
> |  | Ours (Ori) | 52.43 | 4443 | Be succinct, no redundancy!  |
> |  | Ours (Ori) | 52.53 | 3678 | Keep response brief and tight! |
> |  | Ours (Ori) | 54.33 | 4613 | Prioritize brevity, avoid verbosity! |
>
> ### **W3. Analysis of accuracy drop**
>
> - Firstly, We would like to clarify that **ConciseHint does not generally lead to a performance drop**. As shown in **Table 1**, ConciseHint maintains accuracy across most settings and even *improves* performance in several cases. For instance, on Qwen3-8B evaluated on AIME, ConciseHint improves the accuracy of the base model by **+2.66 points**. Additional results in “W1” and “W2” also show that our method maintains the accuracy well.
> - Regarding **ConciseHint-T**, its goal is **aggressive compression**, so a certain reduction in accuracy is expected due to the inherent *efficiency–accuracy trade-off*. Importantly, this trade-off is  **controllable** through the parameter **γ**, which modulates the strength of compression. As shown in **Table 2**, while γ=1.0 yields a more noticeable decrease in accuracy, setting **γ=0.7** maintains a substantial reduction in generation length *with only a modest and acceptable accuracy drop*. This demonstrates that users can flexibly choose the desired balance between efficiency and performance.
>
> **Thank you for your reading. We hope the additional experiments could enhance our paper.**

---

> > ### Comment · Reviewer_Hd4V · 2025-11-28
> >
> > Thank you to the authors for their prompt response and for considering our suggestions. However, I did not find any revisions addressing the suggestions in the manuscript. Could you please confirm whether a new version has been uploaded?

---

> > > ### Author Response · Authors · 2025-12-02
> > >
> > > Thank you for your reply. We have uploaded the revised pdf and marked the changed content in blue.

---

### Official Review · Reviewer_UfHM · 2025-11-01

**Soundness:** 3
**Presentation:** 2
**Contribution:** 2
**Rating:** 4
**Confidence:** 3

**Summary:**

The authors propose ConciseHint to tackle the inefficiency of reasoning models. This is a framework that injects "concise hints" (like "make answer concise!") during the reasoning process, rather than only prompting before it. The method’s key features include: (1) Complexity-Adaptive Intensity, which automatically adjusts how often it injects hints. (2) Dynamic Injection Position, which dynamically adjusts where it injects the hint in the text. A trained version, ConciseHint-T, learns hint embeddings from data to further improve efficiency. Experiments show ConciseHint significantly reduces token usage (e.g., ~49% on GSM8K) while maintaining accuracy.

**Strengths:**

- The "in-reasoning intervention" paradigm is new and interesting; it is intelligently designed to avoid hurting performance.
- The method is flexible and can be integrated with other existing efficiency methods, and it can also be controlled either in a training-free or a trained manner.
- Experimental results show that the method works effectively across multiple state-of-the-art models (Qwen3 series, DeepSeek-R1) and challenging benchmarks.

**Weaknesses:**

- The core assumption relies on the idea that the current reasoning length is a good proxy for query complexity. This largely depends on specific models, as a model can be verbose on an easy problem or concise on a hard one.
- The evaluation methodology is weak:
  - The paper is missing comparisons to other efficient reasoning methods like AlphaOne, AdaptThink, O1-pruner and Autol2s.
  - Missing multiple runs and pass@1: For small, complex benchmarks like AIME24 (only 30 problems), reporting "accuracy" from a single run or small average is not statistically sound, especially when using sampling (temp 0.6).
- The trained version, ConciseHint-T, is trained only on the GSM8K (math) dataset. The paper's claim that it "generalizes well" to completely different domains like GPQA has limited evidence.The hyperparameters are sensitive and need careful choice.
- The paper claims $\alpha$ and $\beta$ work well when fixed, but the appendix shows that poor choices can "significantly undermine accuracy."

[1] AlphaOne: Reasoning Models Thinking Slow and Fast at Test Time

[2] Adaptthink: Reasoning models can learn when to think

[3] O1-pruner: Length-harmonizing fine-tuning for o1-like reasoning pruning

[4] Autol2s: Auto long-short reasoning for efficient large language models

**Questions:**

- Could fine-tuning the model (even with a lightweight method like LoRA) on the same concise dataset also work?
- What is the end-to-end latency impact of the periodic KV-cache invalidation?
- Are there any other proxies for complexity besides current length?

---

> ### Author Response · Authors · 2025-11-24
> **Response to Reviewer #UfHM, Part1**
>
> **Thank you for the constructive comments and suggestions. We respond to your concerns point by point as follows, and are glad to discuss with you.**
>
> ### **W1. Use the current length as the complexity evaluator.**
>
> Following prior work [1,2], we believe the current output length is a good proxy for query difficulty. To understand this point, we need to distinguish **model-perceived difficulty** **from** **objective** **difficulty.**  If a question is objectively easy but the model produces verbose reasoning, that verbosity signals the question is difficult for that model, so adapting hint strength to output length of the model is appropriate. A question might be easy for model A but difficult for model B, so using a fixed objective difficulty (e.g., human annotation) is not proper. Ablation study (L418) demonstrating the effectiveness of our complexity-adaptive strategy based on the length proxy, supporting our conclusion.
>
> ### **W2. The evaluation methodology**
>
> - **Additional baselines.** According to the reviewer’s suggestion, we add comparisons to additional baselines **AlphaOne[3]** and **O1-pruner[4]**. The results on DeepSeek-R1-Distill-Qwen-7B are shown in the following tables. Across all benchmarks, our method consistently outperforms these baselines.
>
> | Dataset | Method | Acc | #Token |
> | --- | --- | --- | --- |
> | AIME24 | Ori. | 46.7 | 6648 |
> |  | alpha-one | 50.0 | 6827 |
> |  | Ours (Ori)  | 50.0 | 6155 |
> |  |  |  |  |
> | AMC23 | Ori. | 82.5 | 4624  |
> |  | alpha-one | 90.0 | 4397 |
> |  | Ours (Ori)  | 90.0 | 3953 |
> |  |  |  |  |
> | MATH-500 | Ori. | 87.6 | 3239 |
> |  | alpha-one | 91.2 | 4337 |
> |  | Ours (Ori)  | 91.0 | 2630 |
>
> | Dataset | Method | Acc | #Token |
> | --- | --- | --- | --- |
> | AIME24 | Ori. | 53.5 | 10306 |
> |  | O1-Pruner | 49.2 | 9719 |
> |  | Ours (Ori) | 57.7 | 8853 |
> |  |  |  |  |
> | GSM8K | Ori. | 87.9  | 682 |
> |  | O1-Pruner | 87.6 | 428 |
> |  | Ours (Ori) | 88.0 | 400 |
> |  |  |  |  |
> | MATH-500 | Ori. | 90.2 | 3674 |
> |  | O1-Pruner | 86.6 | 2534 |
> |  | Ours (Ori) | 90.8 | 2658 |
>
> - **Missing multiple runs.** We apologize for not clearly stating this in the main paper. **All results reported in the submission are averaged over multiple runs.** Specifically, we run each experiment **5 times on GSM8K** and **10 times on AIME and GPQA.** We will explicitly clarify this in the paper to avoid confusion.
>
> We hope these clarifications and additional comparisons address the reviewer’s concerns about evaluation rigor.
>
> ### **W3. Hyperparameters sensitivity of the trained version ConciseHint-T.**
>
> Our claim of generalization means that ConciseHint-T does not learn domain-specific knowledge, but learns a **domain-agnostic pattern of efficient reasoning**. So, hint learned on dataset A can be applied in dataset B. **Table 2** shows that the GSM8K-trained hint embedding transfers effectively to AIME and GPQA**.**  For the selection of hyperparameters, we set γ to control the efficiency-accuracy trade-off. Generally, as shown in **Figure 3**, a larger γ results in more aggressive compression (fewer tokens) but may lead to a larger accuracy drop. γ exposes a *user-controllable knob* that allows practitioners to select the desired compression ratio depending on deployment constraints. The trained version aims to aggressively reduce the token usage. Importantly, when accuracy is the primary concern, users can simply use the training-free ConciseHint, which effectively reduces tokens while maintaining the accuracy well  (Table 1).
>
> ### **W4. The hyperparameters  *α and β***
>
> Our claim that *“α and β work well when fixed”* refers specifically to using **fixed values within the recommended range**, not that arbitrary choices always work. In the main paper, all primary experiments consistently use α = 128 and β = 0.2, which we found to be robust across datasets and models. **We have conducted extensive hyperparameter studies and provided concrete selection guidelines in Section A.1.  Specifically**,
>
> - **Effect of α.** A larger α increases the number of output tokens, typically improving accuracy.   Excessively small α (≤64) harms the accuracy. We therefore recommend **α to be set to a small but not excessively small value, e.g, 128, 256** for a good efficiency-accuracy balance (L683).
> - **Effect of β.** When β < 0.1 (especially β = 0), the accuracy drops noticeably. Hence, we recommend **β ≥ 0.2** (L661) for stable performance.
>
> These analyses were included precisely to help users avoid the poor configurations highlighted by the reviewer. With the recommended ranges, α and β can effectively enhance efficiency without an accuracy drop.

---

> ### Author Response · Authors · 2025-11-24
> **Response to Reviewer #UfHM, Part2**
>
> Q1. Could fine-tuning the model (even with a lightweight method like LoRA) on the same concise dataset also work?
>
> - Directly fine-tuning the model weights on concise data works to some extent. As we discuss in the related works, it belongs to before-reasoning optimization, which is orthogonal to our in-reasoning intervention method.
>
> Q2. What is the end-to-end latency impact of the periodic KV-cache invalidation?
>
> - Section A.2 analyzes the end-to-end latency of our method. Figure 6 shows the relative extra latency caused by KV-cache invalidation is low. Besides, Figure 7 shows the comparison of overall end-to-end latency, which demonstrates that our method significantly reduces the actual reasoning latency.
>
> Q3. Are there any other proxies for complexity besides current length?
>
> - There are ways like using a external model to measure the complexity. But it is hard to precisely measure a problem’s difficulty for a reasoning model.
>
> **Thank you for your reading. We hope the additional experiments could enhance our paper.**
>
> Reference:
>
> [1]  s1: Simple test-time scaling.
>
> [2] How well do llms compress their own chain-of-thought? a token complexity approach
>
> [3] AlphaOne: Reasoning Models Thinking Slow and Fast at Test Time
>
> [4]  O1-pruner: Length-harmonizing fine-tuning for o1-like reasoning pruning

---

### Official Review · Reviewer_LqLF · 2025-11-03

**Soundness:** 2
**Presentation:** 3
**Contribution:** 2
**Rating:** 4
**Confidence:** 4

**Summary:**

This paper focuses on the reasoning efficiency problem of large reasoning models (LRMs) with CoT that tend to generate verbose intermediate reasoning steps. To improve the efficiency, the paper proposes ConciseHint, a method that performs intervention during the generation of reasoning, making the reasoning process concise. ConciseHint continuously injects hints (either a designed text or continuous embeddings) to control the subsequent token generation. ConciseHint also adaptively adjust the injection intensity according to the complexity of the query, which balances efficiency-accuracy by applying a lower hint intensity for complex queries and a higher intensity for easy ones.

**Strengths:**

-	The approach of inserting a short, instructive hint (e.g., “make answer concise”) into the model’s reasoning process is simple and straightforward.
-	The strategy for adjusting the hint injection intervals and positions is intuitive and well-motivated.
-	The paper is clearly written and logically structured.

**Weaknesses:**

-	Limited evaluation. The experiments are run on only three datasets, and two of them (AIME24 and GPQA-Diamond) are quite small. It would be helpful to test the method on more datasets from different domains.
-	Performance drop. While ConciseHint successfully reduces the number of generated tokens, it also causes a clear drop in accuracy, especially for ConciseHint-T.
-	Narrow analysis. The evaluation mainly looks at accuracy and token count. It would be valuable to also assess the quality of the reasoning steps. It’s unclear how the injected hints change the reasoning path, whether they improve clarity, oversimplify the reasoning, or disrupt its flow. A more detailed quality or behavioral analysis would make the work stronger.
-	Limited novelty. Although the method is simple and practical, its scientific novelty is modest. The hint scheduling and injection mechanisms are relatively intuitive and heuristic, and the paper does not analyze why or how the hints affect the model’s reasoning process. Exploring how sensitive the model is to different hint designs or injection positions would add important insights.

**Questions:**

The paper does not discuss the generalizability of the designed hint. Would the same hint work effectively across other reasoning domains, or would they require task-specific tuning?

---

> ### Author Response · Authors · 2025-11-24
> **Response to Reviewer #LqLF, Part 1**
>
> **Thank you for the constructive comments and suggestions. We respond to your concerns point by point as follows  and are glad to discuss with you.**
>
> ### **W1 & Q1.  Evaluation domain.**
>
> Following your suggestion, we have conducted additional experiments on **code generation** and **commonsense reasoning** tasks to further validate the broad applicability of our approach. The results of Qwen3-4B are shown in the following tables:
>
> | Dataset | Method | pass@1 | pass@10 | #Token |
> | --- | --- | --- | --- | --- |
> | Code generation-Human Eval [1] | Ori. | 93.96 | 98.78 | 2715 |
> |  | Ours (Ori) | 94.20 | 96.34 | 2355 |
> |  | Prompt | 93.35 | 96.95 | 2688 |
> |  | Ours (Prompt) | 93.07 | 97.56 | 1983  |
> |  | Deer | 93.41 | 96.95 | 2685 |
> |  | Ours (Deer) | 93.12 | 96.34 | 2218  |
>
> | Commonsense reasoning-CSQA[2] | Method | Acc | #Token |
> | --- | --- | --- | --- |
> |  | Ori. | 81.00 | 657 |
> |  | Ours (Ori) | 80.81 | 463 |
> |  | Prompt | 80.59 | 485 |
> |  | Ours (Prompt) | 80.30 | 408 |
> |  | Deer | 80.11 | 410 |
> |  | Ours (Deer) | 80.00 | 378 |
>
> These results show the efficacy of our method to facilitate efficient reasoning on code generation and commonsense reasoning.
>
> ### **W2.  Performance drop**
>
> - We would like to clarify that **ConciseHint does not generally lead to a performance drop**. As shown in **Table 1**, ConciseHint maintains accuracy across most settings and even *improves* performance in several cases. For instance, on Qwen3-8B evaluated on AIME, ConciseHint improves the accuracy of the base model by **+2.66 points**. Additional results in “W1 & Q1” and “W4” also show that our method maintains the accuracy well.
> - Regarding **ConciseHint-T**, its goal is **aggressive compression**, so a certain reduction in accuracy is expected due to the inherent *efficiency–accuracy trade-off*. Importantly, this trade-off is **controllable** through the parameter γ (Figure 3), which modulates the strength of compression. As shown in Table 2, while γ=1.0 yields a more noticeable decrease in accuracy, setting **γ=0.7** maintains a substantial reduction in generation length *with only a modest and acceptable accuracy drop*. This demonstrates that users can flexibly choose the desired balance between efficiency and performance.
>
> ### **W3.  The influence of our hint on the quality of reasoning chains**
>
> Thank you for your suggestion. We conduct additional quantitative and behavioral analyses to address this concern.
>
> - **Statistics analysis of reasoning transitions.** As shown in Section 4.4 in the original paper, we analyze how ConciseHint changes *transition words* (e.g., *wait*, *alternatively*), which are strong indicators of self-reflection and reasoning refinement. Our results show that while ConciseHint reduces the *number* of transition words, while the *average distance between consecutive transition words* remains unchanged. This indicates that our method removes redundant self-reflection but **does not reduce the depth or structural complexity of each reasoning segment**. In other words, ConciseHint compresses redundancy rather than oversimplifying the reasoning process.
> - **LLM-as-a-judge evaluation of reasoning quality.** To directly assess whether hints change reasoning quality, we conducted an additional pairwise evaluation using GPT-4o-mini as an external judge. Given two reasoning chains (with and without hints) for the same question, the judge was asked: *“Given two reasoning chains, A and B, which one has higher quality, i.e., is more logically coherent, consistent, and structurally sound? Or are they equally good?”* For *correct* responses, the judge labeled the two chains as **equal quality in over 99.5% of cases**. This strongly suggests that ConciseHint does **not** degrade clarity, logical consistency, or structural soundness.
>
> **Together, these analyses show that ConciseHint maintains the quality of the reasoning chain while reducing redundant reflection, resulting in shorter yet equally coherent reasoning paths.**

---

> ### Author Response · Authors · 2025-11-24
> **Response to Reviewer #LqLF, Part2**
>
> ### **W4. Novelty,  How the hints affect the model’s reasoning process, Sensitivity to different hint designs and injection positions**
>
> - **Novelty.** We believe our difficulty-adaptive hint scheduling strategy and dynamic injection strategy are novel. Firstly, our method explores the largely unexplored in-reasoning intervention efficiency improvement. Secondly, the hint scheduling strategy is not heuristic, which is based on our crafted adaptive complexity-based scheduling. The dynamic injection is designed to balance extra costs, to ensure they are negligible, which is supported by detailed cost analysis with formulation and empirical results (Sec. A.2).  Thirdly, we also introduce hint learning besides the manually designed hints. Our extensive main results, ablation results, and additional results demonstrate the effectiveness and necessity of our components.  We think that practicability and simplicity are not grounds for claiming a method lacks novelty. We kindly request the reviewer to recognize the significance of our method.
> - **How the hints affect the model’s reasoning process.** Please refer to the response to W3.
> - **Sensitivity to different hint designs.** Following the reviewer’s suggestion, we additionally conduct a **sensitivity study on different hint designs**, as shown in the following table. Different hint designs consistently reduce the token usage and maintain the accuracy well.
> - **Sensitivity to different injection positions.** We clarify that the paper **already analyzes injection-position sensitivity**: our ablation study (Table 4) shows the effectiveness of proposed dynamic injection strategy compared to different fixed-position injections. **Figure 8** provides concrete cases studies explaining why injection in the tail harms the accuracy.
>
> | Dataset | Method | Acc | #Token | Hint design |
> | --- | --- | --- | --- | --- |
> | GSM8K | Ori. | 94.81 | 2381 | - |
> |  | Ours (Ori) | 94.74 | 1213 | make answer concise! |
> |  | Ours (Ori) | 94.89 | 1296 | Be succinct, no redundancy!  |
> |  | Ours (Ori) | 94.71 | 1092 | Keep response brief and tight! |
> |  | Ours (Ori) | 95.07 | 1366 | Prioritize brevity, avoid verbosity! |
> |  |  |  |  |  |
> | AIME24 | Ori. | 64.33  | 11634 | - |
> |  | Ours (Ori) | 66.67  | 10523 | make answer concise! |
> |  | Ours (Ori) | 66.67 | 10785 | Be succinct, no redundancy!  |
> |  | Ours (Ori) | 66.67 | 9311 | Keep response brief and tight! |
> |  | Ours (Ori) | 65.00 | 10867 | Prioritize brevity, avoid verbosity! |
> |  |  |  |  |  |
> | GPQA | Ori. | 51.82 | 7399 | - |
> |  | Ours (Ori) | 52.73 | 4099 | make answer concise! |
> |  | Ours (Ori) | 52.43 | 4443 | Be succinct, no redundancy!  |
> |  | Ours (Ori) | 52.53 | 3678 | Keep response brief and tight! |
> |  | Ours (Ori) | 54.33 | 4613 | Prioritize brevity, avoid verbosity! |
>
> **Thank you for your reading. We hope the additional experiments could enhance our paper.**
>
> Reference:
>
> [1] Evaluating Large Language Models Trained on Code
>
> [2] CommonsenseQA: A Question Answering Challenge Targeting Commonsense Knowledge

---

### Author Response · Authors · 2025-12-02
**Summary of rebuttal process by authors, Part2**

### **For Reviewer Hd4V:**

1. **W1. Test on larger model**

Following the reviewer’s suggestion, we add experiments on a larger model.

For details, please refer to Response to Reviewer  #Hd4V, W1.

2. **W2. Redundancy Targeting, Parameter Sensitivity Analysis, and Hint Content Impact**
- **Redundancy Targeting.**  Section 4.4 and Table 5 shows that our method reduces a large proportion of redundant transition words like “wait” and “alternatively”, thereby promoting efficient self-reflections and making the overall reasoning more concise.
- **Parameter Sensitivity Analysis.** The reviewer seems to **miss the part of parameter sensitivity analysis in the appendix A.1** of the original paper.  We clarify this.
- **Hint Content Impact.** Following the reviewer’s suggestion, we additionally conduct a **study on the impact of different hint designs.** Different hint designs consistently reduce the token usage and maintain the accuracy well.

For details, please refer to Response to Reviewer  #Hd4V, W2.

3. **W3. Analysis of Performance drop.**

Firstly,  we would like to clarify that generally ConciseHint does **not  lead to a performance drop**. Secondly, for ConciseHint-T (aggressive compression), the efficiency-accuracy trade-off is **controllable** via parameter γ. When setting γ to a lower value (e.g., 0.7), the accuracy drop is acceptable. A certain reduction in accuracy is expected in the case of extreme compression, due to the inherent efficiency–accuracy trade-off.

For details, please refer to Response to Reviewer  #Hd4V, W3.


---------------
### **For Reviewer PxsQ:**

1. **W1 & 2. Hyperparameter Selection and Clarity; Overreliance on Hyperparameters**

**We clarify we have conducted extensive hyperparameter sensitivity experiments and provide clear selection guidelines for users. The reviewer seems to neglect the parameter sensitivity analysis about α and β in the appendix A.1, and the analysis about γ (Figure 3).**

For details, please refer to Response to Reviewer #PxsQ, Part1, W1 & 2.

2. **W3. Incomplete Experimental Coverage**
- Following the reviewer’s suggestion, we conduct the missing baseline comparisons for DeepSeek-R1-14B, strengthening the overall experimental completeness.
- We clarify that we have tested combining prompting-based baselines with ConciseHint, and found it effective.

For details, please refer to Response to Reviewer #PxsQ, Part1, W3.

3. **W4. Overclaim on Reasoning**

Following the reviewer’s suggestion, we conduct additional experiments on **code generation** and **commonsense reasoning** tasks to further validate the broad applicability of our approach.

For details, please refer to Response to Reviewer #PxsQ, Part1, W4.

4. **W5.  Discussion about the Trade-off Between Accuracy and Efficiency**

We clarify the accuracy-efficiency tradeoff.

For details, please refer to Response to Reviewer #PxsQ, Part1, W5.

---------

A Note: We note that three reviewers **may have overlooked our detailed analysis of hyperparameter sensitivity and the corresponding specific selection guidelines provided in Appendix Section A.1,** despite our explicit recommendation in the main text for readers to refer to Appendix A.1. We believe this is the primary reason behind the reviewers’ concerns regarding hyperparameters.

---------

**We upload the corresponding revised pdf, with the changed content marked in blue.**

---

### Author Response · Authors · 2025-12-02
**Summary of rebuttal process by authors, Part1**

## Summary of rebuttal process by authors

We hereby summarize the key idea of reviewer’s concerns and our corresponding responses. We sincerely hope the summary can help AC understand the whole rebuttal process.

### **For Reviewer LqLF:**

1. **W1 & Q1.  Limited evaluation domain**

We supplemented experiments on **code generation** and **commonsense reasoning** tasks to verify generalizability. The results confirm that our method maintains accuracy while reducing tokens across these domains.

For details, please refer to Response to Reviewer #LqLF, Part 1, W1 & Q1.

2. **W2. Performance drop**

Firstly,  we would like to clarify that generally ConciseHint does **not  lead to a performance drop**. Secondly, for ConciseHint-T (aggressive compression), the efficiency-accuracy trade-off is **controllable** via parameter γ. When setting γ to a lower value (e.g., 0.7), the accuracy drop is acceptable. Moreover, additional results in rebuttal also show that our method maintains the accuracy well.

For details, please refer to Response to Reviewer  #LqLF, Part 1, W2.

3. **W3. Lack of analysis of reasoning quality and behaviors.**
- In fact, in Section 4.4 of the original paper, we **have analyzed** the impact of our method on reasoning behaviors based on the statistics analysis of reasoning transition words.
- As supplementary, we conduct LLM-as-a-judge evaluation of reasoning quality, showing our method does not degrade clarity, logical consistency, or structural soundness.

For details, please refer to Response to Reviewer  #LqLF, Part 1, W3.

4. **W4. Novelty, How the hints affect the model’s reasoning process, Sensitivity to different hint designs and injection positions.**
- **Novelty.** We clarify the novelty and significance of our method.
- How the hints affect the model’s reasoning process.  Same as W3.
- **Sensitivity to different hint designs.** Following the reviewer’s suggestion, we additionally conduct a **study on the impact of different hint designs.** Different hint designs consistently reduce the token usage and maintain the accuracy well.
- **Sensitivity to different injection positions.**  We clarify that the paper **has already analyzed** injection-position sensitivity.

For details, please refer to Response to Reviewer  #LqLF, Part 1, W4.

---------------
### **For Reviewer UfHM:**

1. **W1. Use the current length as the complexity evaluator**

We clarify the current output length is a good metric to measure the complexity of the query. We hope the reviewer can **distinguish model-perceived difficulty from the objective difficulty.**

For details, please refer to Response to Reviewer  #UfHM, Part 1, W1.

2. **W2. The evaluation methodology**
- **Additional baselines.** Following the reviewer’s suggestion, we add comparisons to additional baselines AlphaOne and O1-pruner.
- **Missing multiple runs. The reviewer must miss some sentences in our paper. W**e clarify we have pointed out we run multiple times and compute the average in the original paper for all experiments. We make it clearer by clarifying the specific number of run times.

For details, please refer to Response to Reviewer  #UfHM, Part 1, W2.

3. **W3. Hyperparameters sensitivity of the trained version ConciseHint-T.**

We clarify that the “generalize” denotes that the trained hint embeddings indeed learns a domain-agnostic pattern of efficient reasoning, rather than domain-specific knowledge. And the parameter γ is designed to control the compression-accuracy tradeoff.

For details, please refer to Response to Reviewer  #UfHM, Part 1, W3.

4. **W4. The hyperparameters α and β**

We must clarify that  “α and β work well when fixed” refers to using fixed values within the recommended range, not that arbitrary choices. In the original paper, we have conducted **extensive hyperparameters sensitivity studies, and provided clear selection guidelines for users.**

For details, please refer to Response to Reviewer  #UfHM, Part 1, W4.

---

### Meta-Review · Area_Chair_42nE · 2026-01-12

**Summary:**

This paper proposes ConciseHint, an inference-time intervention framework to improve reasoning efficiency by continuously injecting concise hints (either a fixed text hint or learned hint embeddings) while the model is generating chain-of-thought. The method uses a length-based proxy for “complexity” to adapt hint intensity and a dynamic injection-position policy to limit prefilling overhead. The rebuttal adds extra experiments and clarifications, but overall the review discussion remains borderline with persistent concerns about novelty/positioning and baseline coverage.

**Reviewer Concerns:**

**what the authors have added/addressed**:
the authors expanded evaluation beyond the original math-heavy setup (added code + commonsense), added a larger model, clarified that results are averaged over multiple runs, provided additional baselines (e.g., AlphaOne / O1-pruner), and included more sensitivity studies (hint design, hyperparameters) plus some analysis of latency overhead and reasoning quality.

**what remain unaddressed**
1. the novelty claim that prior work “does not intervene during reasoning” is not fully convincing, and the related-work framing is not fair/accurate (some cited methods are not simply “early exit” but indeed do very similar inference-time intervention/probing), and there have been a body of work doing inference-time intervention that this paper should have discussed.

2. Given the existence of inference-time intervention/probing lines of work, the paper still lacks direct comparisons to these stronger, closer baselines, making it hard to assess whether the contribution is more than a prompt-engineering variant.

3. The approach relies on several hyperparameters and on a model-dependent “current length = difficulty” proxy; while the authors provide guidelines, this still raises robustness/generalizability concerns. Also a discussion or proof-of-concept how this could be used in inference/serving systems would make this paper much more appciated.

**Reviewer Scores:**

Given the additional experiments/clarifications, maybe we would see small increase at most for the borderline reviewers who focused on experimental completeness and rigor. That said, I do not expect full discussion to flip the overall consensus to a confident accept because the remaining issues are about positioning/novelty and missing closest baselines, not minor experimental gaps.

---

### Decision · Program_Chairs · 2026-01-26

Reject